# Design of a Novel Wideband Leaf-Shaped Printed Dipole Array Antenna Using a Parasitic Loop for High-Power Jamming Applications

**DOI:** 10.3390/s21206882

**Published:** 2021-10-17

**Authors:** Eunjung Kang, Tae Heung Lim, Seulgi Park, Hosung Choo

**Affiliations:** 1Department of Electronic and Electrical Engineering, Hongik University, Seoul 04066, Korea; ej0901@mail.hongik.ac.kr (E.K.); hschoo@hongik.ac.kr (H.C.); 2Hanwha Systems Co., Ltd., Seongnam City 13524, Korea; sg0212.park@hanwha.com

**Keywords:** jammer antenna, printed dipole, wideband antenna, parasitic element

## Abstract

This paper proposes a novel wideband leaf-shaped printed dipole antenna sensor that uses a parasitic element to improve the impedance matching bandwidth characteristics for high-power jamming applications. The proposed antenna sensor consists of leaf-shaped dipole radiators, matching posts, rectangular slots, and a parasitic loop element. The leaf-shaped dipole radiators are designed with exponential curves to obtain a high directive pattern and are printed on a TLY-5 substrate for high-power durability. The matching posts, rectangular slots, and a parasitic loop element are used to enhance the impedance matching characteristics. The proposed antenna sensor has a measured fractional bandwidth of 66.7% at a center frequency of 4.5 GHz. To confirm the array antenna sensor characteristics, such as its active reflection coefficients (ARCs) and beam steering gains, the proposed single antenna sensor is extended to an 11 × 1 uniform linear array. The average values of the simulated and measured ARCs from 4.5 to 6 GHz are −13.4 dB and −14.7 dB. In addition, the measured bore-sight array gains of the co-polarization are 13.4 dBi and 13.7 dBi at 4 GHz and 5 GHz, while those of the cross-polarizations are −4.9 dBi and −3.4 dBi, respectively. When the beam is steered at a steering angle, *θ*_0_, of 15°, the maximum measured array gains of the co-polarization are 12.2 dBi and 10.3 dBi at 4 GHz and 5 GHz, respectively.

## 1. Introduction

In electronic warfare, high-power jamming systems have been widely used to impede the radio frequency (RF) signal detection of friendly forces by producing interference signals to jam enemy RF radar systems [1]. However, the development of radar design technology has supported such RF radar systems with multifunctional modes to avoid jamming signals by using various frequency bands. Thus, it is essential for high-power jamming applications to also have wideband characteristics in order for antennas to efficiently interfere with RF radar signals with diverse frequencies. Extensive efforts have been devoted to enhancing the frequency bandwidth of antennas by employing various design structures, such as a Vivaldi antenna with a flared-notched shape [2], a folded patch antenna with shorting pins [3], a horn antenna with a substrate-integrated waveguide [4], and double exponentially tapered slot antennas [5,6]. Although these approaches have achieved the wideband characteristics of a single antenna, the physical antenna size is too large to mount on jamming applications with multiple antenna elements. To overcome this problem, many studies have investigated miniaturizing the antenna size by applying a meander line on a log-periodic dipole antenna (LPDA) [7], a hybrid-type antenna with wideband characteristics antennas, i.e., a horn antenna and Vivaldi antenna [8,9], and a printed LPDA on a high dielectric substrate [10]. However, these techniques still encounter the problems of high cost and a complex fabrication process, despite the antenna sensor size reduction. In addition, more detailed research is required to improve the array antenna sensor characteristics to enable high-power durability for jamming applications.

In this paper, we propose the design of a novel wideband leaf-shaped printed dipole antenna using a parasitic element to improve the impedance matching bandwidth characteristics for high-power jamming applications. The proposed antenna sensor consists of leaf-shaped dipole radiators, matching posts, rectangular slots, and a parasitic loop element. The leaf-shaped dipole radiators are designed by employing exponential curves to obtain a high directivity, and they are printed on a TLY-5 substrate to enable high-power durability. To enhance the impedance matching characteristics, the matching posts are shorted between the radiators and the ground plate [11], and the rectangular slots are inserted at the edges of the radiators. In addition, the parasitic loop element is added to further improve the matching bandwidth by adjusting the loop structure. To verify the feasibility of the proposed antenna sensor, it is measured to observe the antenna characteristics, such as the reflection coefficient, gain, and efficiency according to the parasitic loop. Moreover, the proposed design is extended to an 11×1 uniform linear array antenna sensor for high-power jamming applications to examine the array properties, such as the active reflection coefficients (ARCs), array gain, and beam steering performance. The proposed array antenna sensor has ARCs of less than −10 dB and the measured array gains according to the beam steering angles are similar to those of the simulations. The results confirm that the proposed array is suitable for the high-power jamming applications.

## 2. The Proposed Antenna Design and Performance

Figure 1 illustrates the geometry of the proposed antenna sensor, which is composed of a radiating part and a parasitic element part. The radiating part has three components—leaf-shaped dipole radiators, rectangular slots, and matching posts—as shown in Figure 1a. The radiators are designed with inner and outer exponential leaf-shaped curves of *f*_1_*(z)* and *f*_2_*(z)* to obtain the broadband matching characteristics as follows:(1)$$f_{1}(z)=c_{1}e^{r_{i}(z-(h_{1}-h_{1}))}+c_{2},$$
(2)$$f_{2}(z)=c_{3}e^{r_{o}(z-(h_{1}-h_{2}))}+c_{4},$$
where *c*_1_, *c*_2_, *c*_3_, and *c*_4_ are the function coefficients for the exponential curves [12]. The exponent coefficients of *r_i_* and *r_o_* can determine the high directivity, while the inner and outer slopes of the curves can change the current distributions. For high-power durability, the radiators are printed on a TLY-5 substrate (*ε_r_* = 2.2, tan*δ* = 0.0009 from Taconic) with dimensions of *w*_1_ × *h*_1_ × *t* (width × height × thickness). The melting point and dielectric strength of the substrate are 320~340 °C and 106,023 V/mm, which can endure the high-power applications. The feed line length of *h*_2_ is connected to the radiator with a 50-Ω transmission line, which is fed by an SMA connector at the (*f_x_*, *f_y_*) feeding point. The matching posts are employed and shorted between the lower parts of the radiators and the ground plate. To observe the ground effect, the proposed antenna is simulated by varying the width of the square-shaped ground size; the ground size is determined by 100 mm × 100 mm to have less pattern distortion under some frequencies. The distances of *d_l_* and *d_r_* from the feeding lines to the posts can be adjusted to enhance the low-end frequency matching. The matching posts can adjust the electrical length at the resonant frequency band when asymmetrically changing *d_l_* and *d_r_*. These can also miniaturize the antenna size through shorting the outer curve of the dipole radiator and the ground [13,14]. To enhance the high-end frequency matching, the rectangular slots are etched at the dipole radiator edge, where the number of slots is *N*. Each slot is designed with a width of *s_w_* and a length of *s_l_*, as well as a constant interval of *s_i_*. The slots are located at a distance of *h*_3_ to increase the current path for the high-end frequency. To determine the dimension, *N*, and the *h*_3_ of the slots, we carried out parametric studies according to the slot parameters. For example, when increasing the number and size of the slots, the reactance values decrease in the high-end frequency band over 5 GHz, where the slot acts as the capacitive loading in the radiators. Thus, *N* of 12 is determined to improve the impedance matching in the high-end frequency band. Figure 1b represents the parasitic element part printed on the opposite side of the printed dipole radiators. The parasitic element is designed as a simple loop-shaped patch with a width of *w*_2_, a length of *l*_1_, and a height *l*_2_ from the ground. The proposed parasitic element structure can achieve broad impedance matching characteristics and size miniaturization. This is because the capacitive and inductive reactance of the indirect electromagnetic (EM) couplings between the radiators and the parasitic loop can be adjusted by changing the parameters of the proposed loop structure.

Figure 2a presents the reflection coefficients in accordance with the existence of the antenna elements (the parasitic loop and slots) to observe the effects on the impedance matching characteristics. We simulated the proposed antenna when each element was removed one by one. The resulting −10 dB fractional bandwidths at the center frequency of 4.5 GHz are 66.7% (the proposed antenna), 41.1% (without the parasitic loop), 64.4% (without the slots), and 10.4% (without the slots and parasitic loop). In particular, the stand-alone dipole with the parasitic element drastically enhances the impedance matching characteristics, meaning that we can further analyze the parametric study for the shape of the parasitic loop. Figure 2b illustrates the fractional bandwidth according to the ratio of the parasitic loop thickness and length. The solid and dashed lines indicate the bandwidth results when considering a thickness of *w*_2_ and a loop length of *l*_1_. As the loop length, *l*_1_, increased from 35 mm to 115.6 mm, the fractional bandwidth was gradually enhanced. The wide fractional bandwidth can be obtained with the narrow width of 0.3 mm when *w*_2_ varies from 0.3 mm to 7 mm. This is because the strong mutual coupling strengths between the radiator and the parasitic loop can adjust the input impedance of the proposed antenna in the low- and high-end frequency band. Thus, the maximum fractional bandwidth of 66.7% can be obtained when *l*_1_ is 115.6 mm and *w*_2_ is 0.3 mm. Figure 2c,d show the reflection coefficients in accordance with the slot height, *h*_3_, and the number of the slots, *N*. When changing the slot height, *h*_3_, from 25 mm to 70 mm, the resonant frequency band became down-shifted. This is because the current path in the radiators can be adjusted by the slot height, which affects the resonant frequency band. In addition, the number and size of the slots can tune the reactance values in the high-end frequency band over 5 GHz, where the slot acts as the capacitive loading in the radiators. Through the parametric studies, *N* of 12 and *h*_3_ of 40 mm were determined to improve the impedance matching in the high-end frequency band. The optimized design parameters were obtained using the CST Studio Suite [15], and the detailed values are listed in Table 1. To confirm the feasibility of the proposed antenna sensor, it was fabricated and measured in a full anechoic chamber.

Figure 3 shows photographs of the fabricated antenna sensor printed on the TLY-5 substrate for high-power durability; the proposed antenna sensor is directly fed by an SMA connector. Figure 4 presents the measured and simulated reflection coefficients; it can be seen that the results agree well with each other. The maximum values for the measured and simulated reflection coefficients are −10.1 dB and −10.9 dB from 3 GHz to 6 GHz, respectively. The measured fractional matching bandwidth was improved from 55.6% to 66.7% when the parasitic loop element was applied. Figure 5 presents the simulated and measured maximum gains of the proposed antenna in the absence and presence of the parasitic loop. The solid and dashed lines indicate the gain results of the simulation with and without the parasitic loop, while “o” and “×” markers represent those of the measurements. The enhancement of the maximum gain at 3 GHz is 3 dB, because the parasitic loop element can improve the impedance matching characteristic in the low-end frequency band. With the parasitic loop element, the improvements of the measured and simulated radiation efficiencies at 3 GHz are 30% and 40% due to the enhancement of the matching characteristics in the low-end frequency band. A slight gain discrepancy occurs due to the measurement setup alignments and fabrication errors. Figure 6 shows the simulated and measured radiation patterns of the co- and cross-polarizations in the *zx*- and *zy*-planes. The measurements agree well with the simulations, and the measured half-power beamwidths in the *zx*-plane are 51.2°, 105.2°, and 62° at 3 GHz, 4.5 GHz, and 6 GHz.

## 3. Array Extension and Performance

To examine the array performance for high-power jamming applications, the proposed array element was re-optimized based on the stand-alone antenna in Chapter 2, where a linear periodic structure was used to account for the mutual couplings between the adjacent elements. Note that the width, *w*_1_, of the array element (the same as the array distance) is determined by considering the grating lobe that appears when steering the beam. Herein, we used the odd number, 11, of the array element for the finite array to compare the antenna characteristics of the infinite array as similarly as possible. This was because the odd number could define the exact center element of the finite array. The optimized parameter values of the array element are listed in Table 2. Figure 7a,b show the fabricated 11 × 1 uniform linear array antenna sensor used to confirm the array antenna sensor characteristics. Figure 7c presents a photograph of the measurement setup in a full anechoic chamber. In this setup, the phase center of the array is set to the center element (Port 6), and then the array gain with the beam steering performance is observed by using the phase offset according to the steering angle. For example, when the steering angle is 15°, the array pattern is calculated using a phase difference of 47.5° between the adjacent ports. In high-power jamming applications, it is important to measure the ARCs because all ports of the array antenna sensor are excited with a high power. The ARCs can be calculated based on the equation below [16,17,18]:(3)$$\Gamma_{m}(\theta_{0})=\frac{\sum_{n=1}^{N}S_{mn}e^{-j(n-1)kd\sin\theta_{0}}}{e^{-j(m-1)kd\sin\theta_{0}}}=\sum_{n=1}^{N}S_{mn}e^{-j(n-m)kd\sin\theta_{0}}.$$
where Γ*_m_* is the ARC of the *m*th port and *S_mn_* indicates an *N* × *N* scattering matrix at the *m*th column and *n*th row. *N* is the number of elements and *θ*_0_ is the steering angle. *k* and *d* are the wave number and array distance between the adjacent elements.

Figure 8 shows the comparisons of the simulated and measured ARCs according to the number of array elements. The solid, dashed, dotted, and dash-dotted lines indicate the simulations of the 3 × 1, 5 × 1, 11 × 1, and infinite uniform linear arrays, respectively. The dash-dotted-dotted line represents the measurement of the proposed 11 × 1 linear array antenna sensor. As the number of the array elements increased, the resulting ARCs decreased, especially in the high-end frequency band. Regarding the adjustment of the loop dimension for the periodic structure, the mutual coupling was decreased between the adjacent elements to enhance the ARCs. The average values of the simulated ARCs from 4.5 to 6 GHz are −11.5 dB, −12.2 dB, −13.4 dB, and −16.2 dB for the 3 × 1, 5 × 1, 11 × 1, and infinite linear arrays, respectively, while that of the measurement for the proposed array is −14.7 dB. Figure 9 shows the simulated and measured mutual couplings between the center element (Port 6) and the other elements. The measured and simulated average values of the mutual coupling results are −49.3 dB and −51 dB. We examined the mutual couplings (S_2, 1_, S_3, 1_, S_4, 1_, …, and S_10, 1_) of the 11 × 1 array; the average value of the simulation was −29.8 dB. In addition, the beam steering performance, which is another essential array characteristic for high-power jamming applications, were investigated by measuring the active element patterns (AEPs) of the proposed array. To measure the AEPs, each array element was excited, while the other ports were terminated with 50 Ω loads. Then, the AEPs for all ports were weighted and summed to calculate the steered array gains [19,20]. Herein, we assumed that the feeding network with the phase shifters was well designed with ideal characteristics. The array gain was calculated using the AEPs of all ports based on the following equation:(4)$$P_{array}\left(\theta ,\phi \right)=\frac{\sum_{n=1}^{N}{\bar{w}}_{n}{\bar{v}}_{n}\left(\theta ,\phi \right)}{\sqrt{\sum_{n=1}^{N}{\left|w_{n}\right|}^{2}}},$$
where *v_n_* is a complex vector of the AEP of an *n*^th^ port and *w_n_* is a weighting vector for the beam steering.

Figure 10 shows the beam steering characteristics with steering angles, *θ*_0_, of 0° and 15° at 4 GHz and 5 GHz. The solid and dashed lines indicate the measurements and simulations, and the blue and red lines denote the radiation patterns of the co- and cross-polarization. The measured and simulated results are well matched to each other. The measured bore-sight array gains of the co-polarization are 13.4 dBi and 13.7 dBi at 4 GHz and 5 GHz, and those of the cross-polarizations are −4.9 dBi and −3.4 dBi, respectively. When the beam is steered at the steering angle, *θ*_0_, of 15°, the maximum measured array gains of the co-polarization are 12.2 dBi and 10.3 dBi at 4 GHz and 5 GHz, respectively. Moreover, the co-and cross-polarization level differences at the angle of the maximum gains are 14 dB at 4 GHz and 11.4 dB at 5 GHz. Note that these beam steering results were calculated considering the ideal gain increment of the 11 elements from the bore-sight gains of 3.5 dBi at 4 GHz for the center array element. In addition, the measured back lobe levels seem to be higher than the simulated results because the extra obstacles, such as a RF cable and a positioner structure in the measurement setup, caused high back lobe levels. These results confirm that the proposed array antenna sensor can be applied to high-power jamming applications, as it is capable of achieving essential and required array performances. We also compared the antenna characteristics between the proposed array and the reference wideband arrays; the detailed explanations are listed in Table 3.

## 4. Conclusions

The design of a novel wideband leaf-shaped printed dipole antenna sensor using a parasitic element was proposed to improve the impedance-matching bandwidth characteristics for high-power jamming applications. To obtain the desired wideband characteristics, the proposed antenna sensor was constructed of simple geometry elements: leaf-shaped dipole radiators, matching posts, rectangular slots, and a parasitic loop element. The leaf-shaped dipole radiators were designed with exponential curves and a TLY-5 substrate was employed to achieve high-power durability. The matching posts, rectangular slots, and parasitic elements were used to enhance the impedance matching characteristics. The measured average bore-sight gain and maximum reflection coefficient of the fabricated antenna sensor were 4.3 dBi and −10.1 dB in the frequency band from 3 GHz to 6 GHz. In addition, the fractional bandwidth was 66.7% at the center frequency of 4.5 GHz. The proposed single antenna sensor was re-optimized using a periodic structure as an infinite array, and it was extended to a uniform linear array in order to confirm the array performances, such as the ARCs, AEPs, and beam steering gains. The fabricated 11 × 1 uniform array antenna sensor had an averaged ARC value of −14.7 dB from 4.5 to 6 GHz, while the array gains of the co-polarization at the steering angles of 0° and 15° were 13.7 dBi and 10.3 dBi at 5 GHz.

## Figures and Tables

**Figure 1 sensors-21-06882-f001:**
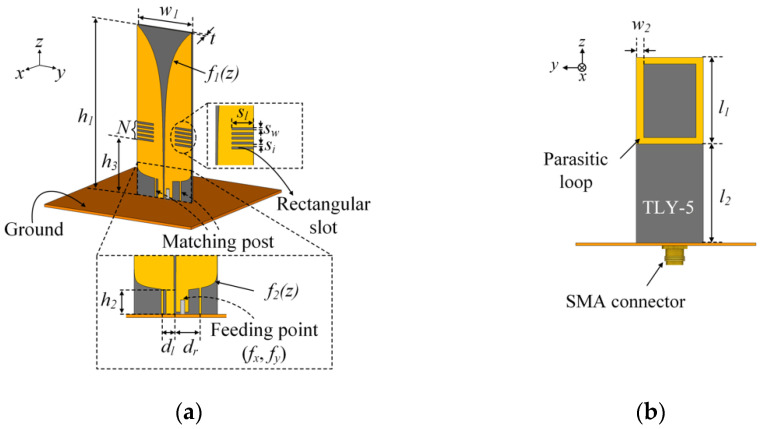
Geometry of the proposed printed dipole antenna: (**a**) isometric view; (**b**) back view.

**Figure 2 sensors-21-06882-f002:**
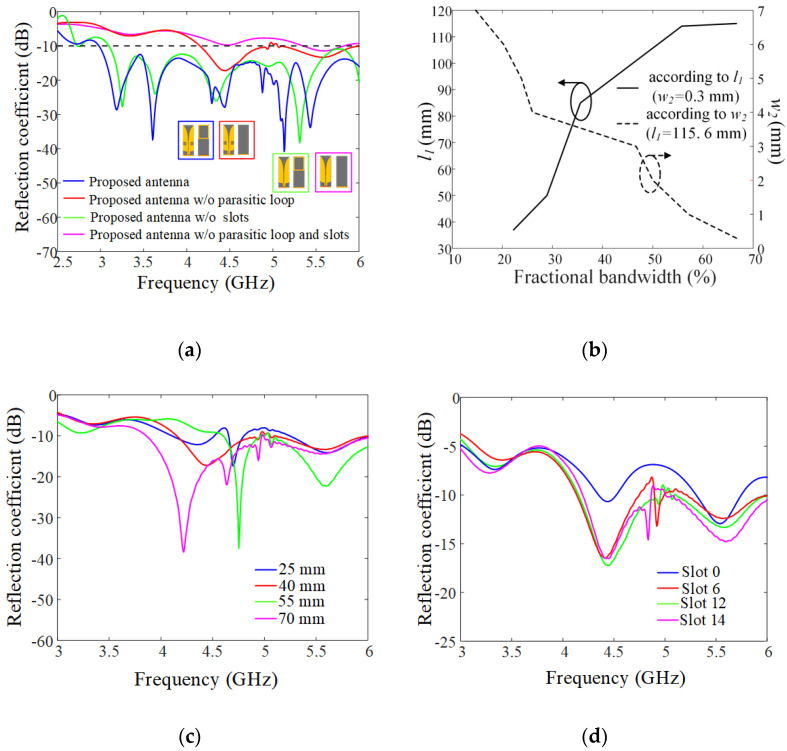
Parametric studies on the rectangular slots and the parasitic loop: (**a**) reflection coefficients according to the antenna sensor components; (**b**) fractional bandwidth according to the parasitic loop structure; (**c**) reflection coefficients according to *h*_3_; (**d**) reflection coefficients according to *N*.

**Figure 3 sensors-21-06882-f003:**
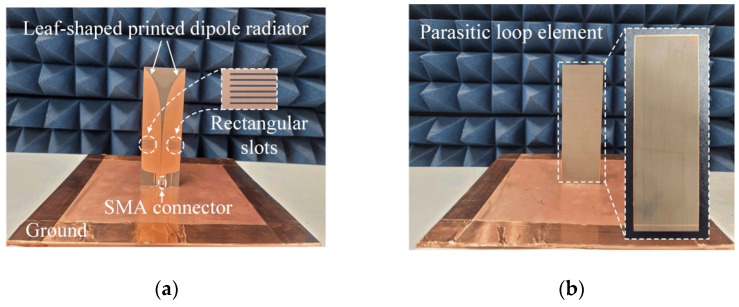
Photographs of the fabricated antenna sensor: (**a**) front view; (**b**) back view.

**Figure 4 sensors-21-06882-f004:**
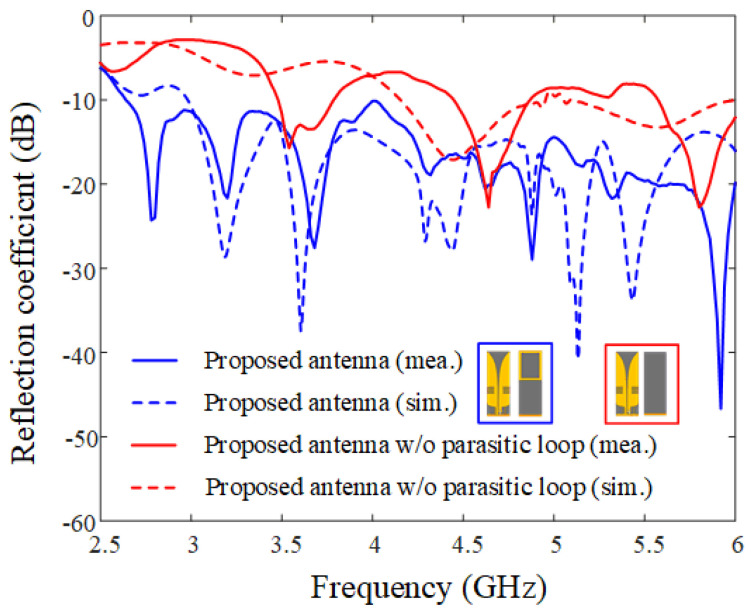
Simulated and measured reflection coefficients of the proposed antenna.

**Figure 5 sensors-21-06882-f005:**
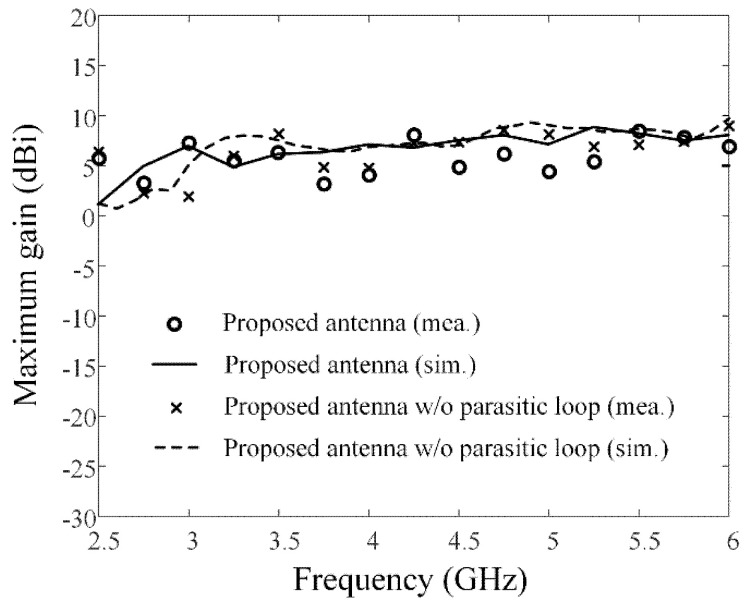
Simulated and measured bore-sight gains of the proposed antenna.

**Figure 6 sensors-21-06882-f006:**
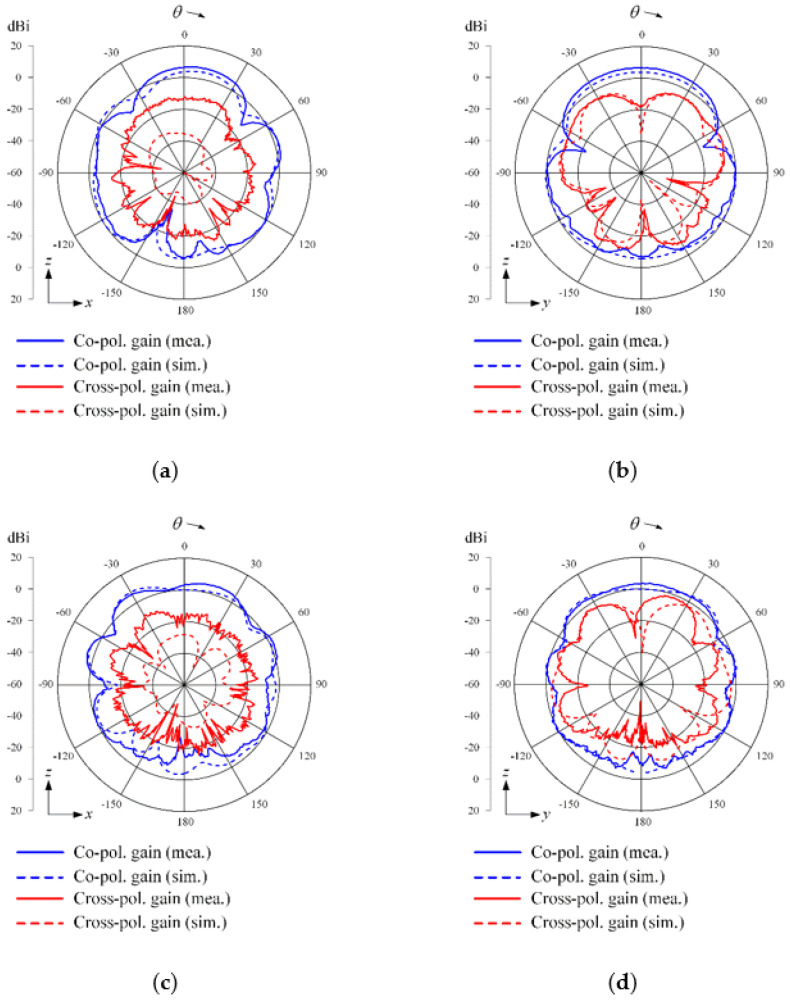
2D radiation patterns: (**a**) *zx*-plane at 3 GHz; (**b**) *zy*-plane at 3 GHz; (**c**) *zx*-plane at 4.5 GHz; (**d**) *zy*-plane at 4.5 GHz; (**e**) *zx*-plane at 6 GHz; (**f**) *zy*-plane at 6 GHz.

**Figure 7 sensors-21-06882-f007:**
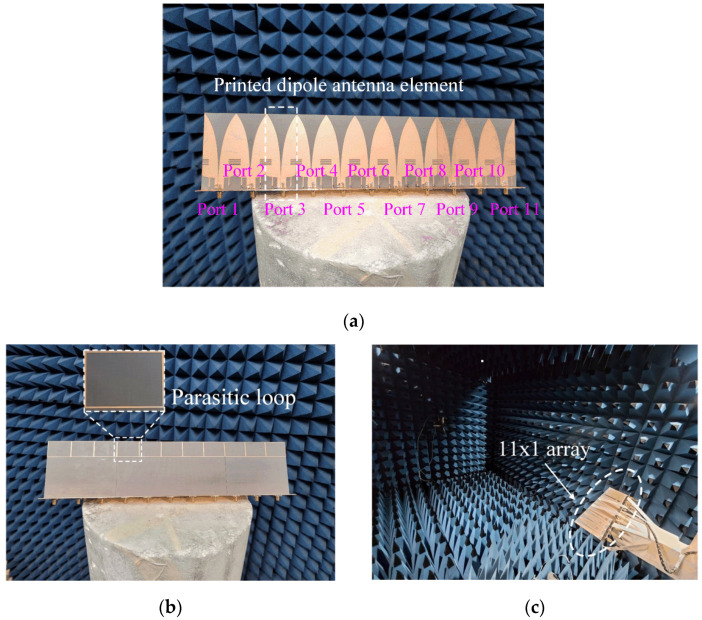
Photographs of the fabricated 11 × 1 uniform linear array antenna and measurement setup: (**a**) front view; (**b**) back view; (**c**) measurement setup.

**Figure 8 sensors-21-06882-f008:**
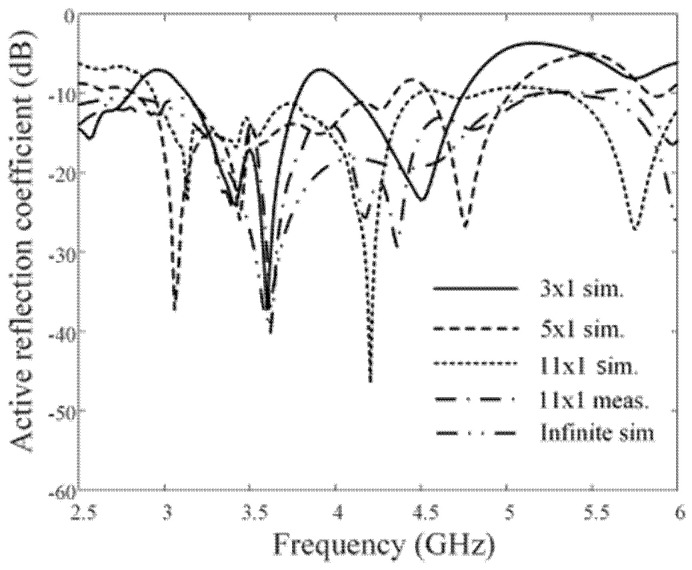
Comparisons of the simulated and measured ARCs according to the number of array elements.

**Figure 9 sensors-21-06882-f009:**
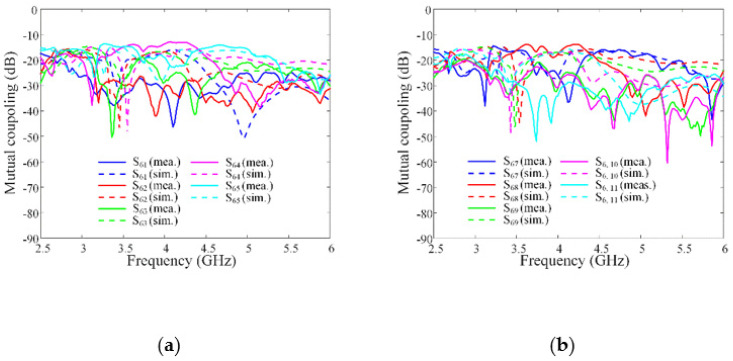
The measured and simulated mutual couplings: (**a**) the mutual couplings for port 1 to port 5; (**b**) the mutual couplings for port 7 to port 11.

**Figure 10 sensors-21-06882-f010:**
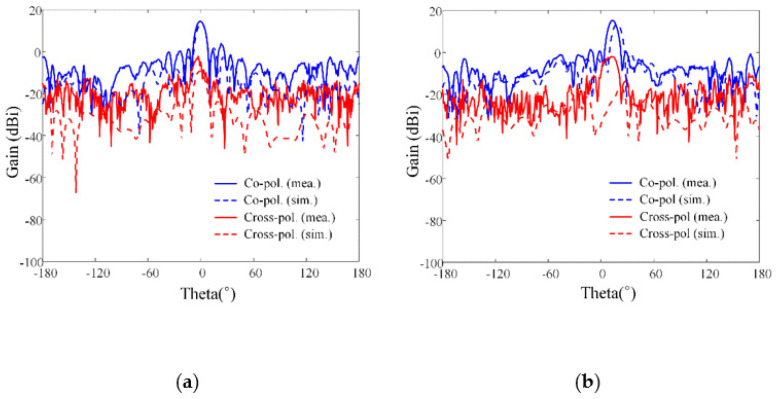
Array beam steering gains at 4 GHz and 5 GHz: (**a**) *θ*_0_ of 0° at 4 GHz; (**b**) *θ*_0_ of 15° at 4 GHz; (**c**) *θ*_0_ of 0° at 5 GHz; (**d**) *θ*_0_ of 15° at 5 GHz.

**Table 1 sensors-21-06882-t001:** Values of the proposed antenna.

Parameter	Values	Parameter	Values
*h* _1_	119 mm	*l* _1_	115.6 mm
*h* _2_	12 mm	*l* _2_	3.4 mm
*h* _3_	40 mm	*d_r_*	30.7 mm
*w* _1_	39.4 mm	*d_l_*	35.4 mm
*w* _2_	0.3 mm	*c* _1_	−0.025
*r_i_*	0.056	*c* _2_	−39.5
*r_o_*	−0.6	*c* _3_	−5.6 × 10^4^
*s* _l_	12 mm	*c* _4_	−15.9
*s_w_*	0.5 mm	*t*	1.6 mm
*s_i_*	1 mm	(*f_x,_ f_y_*)	(0, 22.8)
*N*	5		

**Table 2 sensors-21-06882-t002:** Values of the optimized array antenna element.

Parameter	Values	Parameter	Values
*h* _1_	96.3 mm	*l* _1_	25.8 mm
*h* _2_	14 mm	*l* _2_	93.6 mm
*h* _3_	29.9 mm	*d_r_*	9.2 mm
*w* _1_	38.2 mm	*d_l_*	12 mm
*w* _2_	0.9 mm	*c_1_*	−49.1
*r_i_*	0.038	*c* _2_	−89.9
*r_o_*	−0.9	*c* _3_	−4.5 × 10^6^
*s_l_*	7.3 mm	*c* _4_	−15.3
*s_w_*	0.8 mm	*t*	1.6 mm
*s_i_*	0.6 mm	(*f_x,_ f_y_*)	(0, 22.8)
*N*	5		

**Table 3 sensors-21-06882-t003:** Comparison of the wideband array.

Reference	Array Dimension (Width mm × Length mm × Thickness mm)	Operating Frequency Band (GHz)	The Number of Elements	Substrate Material	Array Gain (dBi)
[9]	500 × 500 × 1001.5	1.75–3	4	Metal	19.7 (at 2.45 GHz)
[12]	480 × 210	2–4	8	Rogers RT5880	≥12 (at 2 to 4 GHz)
[13]	579.12 × 579.12 × 65.6	0.3–2.15	64	TLY-5	20 (at 2 GHz)
[21]	167.48 × 158.25 × 0.6	2.5–6.8 and 7.5–9.5	6	Taconic substrate (*ε_r_* = 4.3, tan*δ* = 0.0035)	14.12 (at 4.5 GHz)
[22]	43 × 72 × 0.762	7–11.5	4	Rogers 3003	12.1 (at 10.7 GHz)
Proposed array	420.2 × 96.3 × 1.6	3–6	11	TLY-5	13.7 (at 4.5 GHz)

## Data Availability

Not applicable.

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
