# Peer review of "Design of a Novel Wideband Leaf-Shaped Printed Dipole Array Antenna Using a Parasitic Loop for High-Power Jamming Applications"

_sensors, 2021, doi:10.3390/s21206882_

Round 1

Reviewer 1 Report

The authors addressed my concerns. 

Reviewer 2 Report

The authors presented a Vivaldi dipole antenna backed with a parasitic element. The Reviewer has some concerns about this manuscript, which are as under.

1. The novelty and contribution of this work should be clearly mentioned in the manuscript.

2. The references and literature review are not enough to locate this work in state-of-the-art. There are many wideband arrays that have been presented even in the MDPI journals, for example, https://doi.org/10.3390/s21186091. Moreover, the addition of a performance comparison table with similar antenna arrays is also recommended.

3. The physical mechanism of the antenna has not been explained in the manuscript. The authors should explain how the slots and parasitic elements contribute to impedance matching. 

4. Figure 2(a) shows the antenna performance for different configurations. According to the Figure, the printed dipole antenna does not resonate (S11 above -10dB across the entire frequency), which is not true. Authors should compare the optimized printed dipole antenna with resonance with the rest of the configurations (antenna with slot and parasitic loop). 

5. The simulated results of the proposed antenna (with parasitic loop) shown in Figure 2(a) and Figure (4) are different.   

6. There is no power divider in the 11x1 array. Each antenna has been fed separately.  The fabricated prototype has no phase shifter as can be seen in Figure 7. How the array was formed and how it was measured without a power divider? 

7. Why does the impedance bandwidth of the array change (Figure 8) significantly with the different number of antenna elements?

Round 2

Reviewer 2 Report

The authors have revised the manuscript well. All concerns of the Reviewer has been addressed.

This manuscript is a resubmission of an earlier submission. The following is a list of the peer review reports and author responses from that submission.

Round 1

Reviewer 1 Report

The authors proposed well designed antenna as the array elements.

However, such array antenna is required the feeding network for example power dividers and other required staff in simulations and measurements (based on sidle-lobes, array feeding

are uniform.

Also, plots for mutual Smn-parameters should be provided. Because, the array without enough isolation (Typically isolation less than 20 dB at operating frequency makes array non useful for practical applications, especially if you want still proposing array as multi-port array without feeding network)

With feeding network, it will be single port design and these effects can be seen in final pattern (Co-Polarization) and cross polarization. Also, cross polar patterns are missing in the figures.

Another issue, for such a array element design, the backlobe of array patterns are unusually 

high.

Reviewer 2 Report

Please see my comments below

  1. The novelty of the paper is very limited. First of all, the antenna configuration (Dual Exponential Tapered Slot Antenna, DETSA) is very mature yet the authors adopted a somewhat "abnormal" way to excite such antenna. It is suggested that the details of the antenna configuration and explanations should be disclosed. The following points need to be clarified:
    (a) the bottom ground plane does have strong effect on the antenna characteristics. There is no discussion about the impact of the ground plane on the matching and radiation characteristics.
    (b) Discussions of the "matching post" is missing. Why would this be necessary? Any published DETSA would achieve better performance using very simple excitation configuration without excess matching components. Please explain and reference DETSA publications adequately.
    (c) The thickness of the substrate is only 1.6 mm which is very small compared to the guided wavelength at the operating frequency. Under such circumferences, the effect of placing a parasitic element in the back of this "end-fire" type of antenna could be doubtful. The authors NEED to show the performance with and without the parasitic elements in terms of the matching, gain, and efficiency from measurement.
    (d) What are the main contributions of the slots cut in the radiator? How do you determine the location, size, and the number of the slots? Please explain.
    (e) For an antenna to be claimed broadband, you NEED to disclose the antenna radiation characteristics as functions of frequencies as well. Please show the measured gain, directivity, and radiation efficiency of the antenna. Radiation pattern should also be disclosed.
    (f)It is very surprising that the measured and simulated gain (Figure 5) differs so much. Also, a severe fluctuation of about 10dB is observed around 3-3.5 GHz and elsewhere. This is a very rare case especially for wideband antennas. Detail and reasonable explanations are required. 
  2. As for array extension, again, there are a lot of details that are not disclosed.
    (a)In general people use even number of antenna elements especially for beam steering purposes. Is there any reason to pick 11 as the number?
    (b)From Figure 6, it seems that the 11 antennas are excited separately. Do you have any feeding network to compose a real "array"? Or this is just 11 antennas placed side by side? 
    (c)Beam steering characteristics were shown in Figure 8 without any details in the setup. How were the phase offset provided to each antenna element?
    (d) Usually the gain of the antenna array increases compared to single antenna element. However, in Line 177 you claimed that the gain is 10.5 dBi at 4 GHz. Compared to the measured gain of roughly 8 dBi in Figure 5 for the single element case, this number is not correct. Please explain why the gain of the array is so low. Usually a 2-element array will give you a 3-dB increase in gain easily, let alone an 11-element one.